# Longleaf Pine Patch Dynamics Influence Ground-Layer Vegetation in Old-Growth Pine Savanna

**Maria Paula Mugnani** [1],*[ID], **Kevin M. Robertson** [2][ID], **Deborah L. Miller** [3][ID] and **William J. Platt** [4][ID]

1  Department of Wildlife Ecology and Conservation, University of Florida, Gainesville, FL 32611, USA
2  Tall Timbers Research Station, Tallahassee, FL 32312, USA; krobertson@talltimbers.org
3  Department of Wildlife Ecology and Conservation, University of Florida West Florida Research and Education Center, Milton, FL 32583, USA; dlmi@ufl.edu
4  Department of Biological Sciences, Louisiana State University, Baton Rouge, LA 70803, USA; btplat@lsu.edu
*  Correspondence: mmugnani23@ufl.edu; Tel.: +1-802-380-9163

**Abstract:** Old-growth longleaf pine savannas are characterized by diverse ground-layer plant communities comprised of graminoids, forbs, and woody plants. These communities co-exist with variable-aged patches containing similar-aged trees of longleaf pine (*Pinus palustris* Mill.). We tested the conceptual model that physical conditions related to the cycle of longleaf pine regeneration (stand structure, soil attributes, fire effects, and light) influence plant species' composition and spatial heterogeneity of ground-layer vegetation. We used a chrono-sequence approach in which local patches represented six stages of the regeneration cycle, from open areas without trees (gaps) to trees several centuries old, based on a 40-year population study and increment cores of trees. We measured soil characteristics, patch stand structure, fuel loads and consumption during fires, plant productivity, and ground-layer plant species composition. Patch characteristics (e.g., tree density, basal diameter, soil carbon, and fire heat release) indicated a cyclical pattern that corresponded to the establishment, growth, and mortality of trees over a period of approximately three centuries. We found that plants in the families Fabaceae and Asteraceae and certain genera were significantly associated with a particular patch stage or ranges of patch stages, presumably responding to changes in physical conditions of patches over time. However, whole-community-level analyses did not indicate associations between the patch stage and distinct plant communities. Our study indicates that changes in composition and the structure of pine patches contribute to patterns in spatial and temporal heterogeneity in physical characteristics, fire regimes, and species composition of the ground-layer vegetation in old-growth pine savanna.

**Keywords:** *Pinus palustris*; fire effects; spatial heterogeneity; plant species niches; plant life history characteristics

## 1. Introduction

Forest, woodland, and savanna communities dominated by a single tree species are often structured as arrays of discrete patches containing even-aged and similar-sized trees [1,2]. Such patches, which reflect spatio-temporal patterns of tree recruitment, growth, and mortality, can simultaneously represent the full suite of stages characterizing the tree life cycle [3,4]. The spatial structure of tree populations organized by such patches, in turn, can influence the species composition of herbs and relative cover by modifying resource availability and growth conditions [5], including soil moisture, nutrient levels, and light availability [6]. Small-scale changes in tree patch dynamics can affect ground-layer plant communities through limitations in growth conditions, which create distinct patchworks of

micro-communities in the understory [7–9]. Such canopy-ground layer interactions are likely in savannas in which the dominant tree species influence fire characteristics and, thus, modify local environments [2,10].

We hypothesized that patch dynamics of longleaf pine (*Pinus palustris* Mill.) influence ground-layer dynamics and herbaceous species composition. Old growth longleaf pine savannas are characterized by an open canopy structure typified by patches containing even-aged clusters of pines established at various times [2,3,11]. The diverse ground-layer plant community contains graminoids, forbs, and broadleaf woody species, with grasses typically the aspect dominants [12–14]. The open savanna condition of old-growth sites is maintained by frequent fire, which prevents dominance of the canopy by woody plants [13,15]. Frequent fire also limits accumulation of fine fuels and duff and therefore results in low severity fires [10,16,17]. The predominantly perennial herbaceous plants in the ground-layer typically survive top killing by fires and quickly re-sprout [16,18,19]. Such frequent fires, especially those that occur in the season of natural lightning fires [20], are characteristic of old-growth longleaf pine savanna [13].

The spatial and age distributions of longleaf pine in old-growth pine savannas are strongly influenced by the species' life history [2]. Longleaf pine cone production is irregular, with periodic mast years separated by years of low cone production [3,21]. Following seed dispersal, which can reach as much as 60 m from large trees [22], newly germinated seedlings are abundant. However, subsequent fires kill many seedlings [23,24]. Even so, seedlings can survive within openings, if needle litter fuel loads are lower and fires are less severe [24–26]. Following mast years, dense patches of even-aged/sized individuals [3,4,26] may be recruited within such openings.

Once formed, patches can persist for at least two or three centuries, as indicated by the increment boring of trees [3,4,27]. During that time, trees grow into the canopy and the initially high density of individuals decreases due to resource competition and mortality from fire [4,11,24]. After trees reach canopy height, mortality from lightning strikes, windstorms, and fires continue to reduce the density of patches over centuries [4,28]. Ultimately, as patches of pines dissipate, open areas of sufficient size for recruitment are formed [26]. In these areas, recruitment following mast years renews the patch cycle as long as fires occur frequently and maintain open areas suitable for regeneration [13,26]. The staggered initiation times of longleaf pine patches in old-growth pine savanna results in patch ages that may span centuries [29].

Environmental characteristics that differ among stages in the patch cycle may influence plant species composition and species richness of the ground-layer vegetation. One potential mechanism by which the longleaf pine patch stage may influence plant species composition is fire effects, as mediated by litter deposition [10]. Pine needle litter in pine savannas often represents approximately half of available fire fuel [10,18,30] but varies with tree age and density, which contributes to patchy fire severity [24,25]. Woody debris, including pinecones, also increases plant mortality beneath pine canopies because of its long combustion residence times [31]. Other environmental characteristics that could vary with the patch stage include light levels [32–34], soil moisture [12,35], and available soil N and C [36,37]. Thus, patch stages could directly or indirectly influence spatial heterogeneity in the ground-layer plant community.

In this study, we test our conceptual model and hypothesize that the longleaf pine (*Pinus palustris*) patch stage and associated environmental characteristics generate spatial heterogeneity of ground-layer plant species composition. We compare physical characteristics and plant species composition among patch stages comprising a chrono-sequence that spans approximately three centuries in an old-growth longleaf pine savanna in southern Georgia, USA. Our results expand the patch dynamics concepts already established for old-growth pine savannas [13], which indicates important relationships between over-story pine patch dynamics and the composition and spatial species distribution of ground layer plant communities.

## 2. Materials and Methods

### 2.1. Study Area and Patches of Longleaf Pine

We conducted our study on the Wade Tract (30°45′N; 84°0′W, Thomas County, Georgia, USA). The site is situated on moderately dissected terrain 25 to 50 m above sea level in the headwaters of the St. Marks River in the Red Hills region of northern Florida-southern Georgia. Soils at the site are loamy-sand Ultisols (Typic and Arenic Kandiudults) [38,39] with clay-rich sub-horizons formed in Pliocene-aged sands of the Miccosukee Formation. This site contains 85 ha of exemplary old-growth pine savanna [13,14,40] protected by a conservation easement held by Tall Timbers Land Conservancy and managed during the past century for northern bobwhite (quail) hunting and conservation.

Much of the ground layer vegetation is dominated by warm season grasses, especially *Aristida beyrichiana* Trinius & Ruprecht, *Schizachyrium scoparium* (Michaux) Nash, and *Sorghastrum secundum* (Elliott) Nash. The plant community is species-rich, with >500 native plant species collected within the Wade Tract easement. Records indicate 27 prescribed fires within each of the two fire management units between 1982 and 2016 (1.5-year average return interval) even though the seasonal timing and often the year of fire differed between units. In addition, 90% of burns were in mid-March to late June. Such high fire frequency is consistent with records from the last three centuries [41,42]. Sufficient fuel loads are easily maintained by substantial annual rainfall (average of ~1350 mm in the vicinity) and a 10–11 month growing season, which results in rapid regrowth of ground-layer plants [14,40].

We identified patches to study within an area covering most of the Wade Tract where the trees were mapped and a census was completed in 1978. At that time, W.J. Platt established a 50-hectare plot within which all trees >1.5 m tall were tagged, mapped, and measured for diameter at breast height. This plot has been re-censused for tree growth, mortality, and recruitment every 3–4 years. We used tree diameter data and increment cores from approximately 400 trees [3,4] to estimate each individual's age based on their diameter. Although tree cores were not cross-dated, we were reasonably confident in patch age designations, given that longleaf pine in the study region have approximately equal amounts of early wood and late wood per ring. This facilitates counting annual rings within the 40–year margin of error represented by the intervals between regenerating cluster age categories [43]. Using ArcGIS 10.5 (ESRI Inc., Redlands, CA), we digitized polygons representing patches identified as clusters of similarly aged trees (indicating concurrent recruitment into the stand). We assigned these patches to one of six categories representing stages of the patch cycle: even-aged clusters of trees aged at 10–50 years, 50–90 years, 90–130 years, 130–180 years, 180–250 years, and open gap areas (Figure A1). Patches of trees were limited to those exceeding 10 m in breadth. Although there were trees considerably older than 250 years [27], patches dissipate into a matrix of scattered large trees as older trees die [3]. Thus, patches transition to open areas without trees over time. Such open areas constituted the majority of the mapped plot (~70% of the total area) when the study was initiated [3]. We identified gaps (an additional stage) as treeless areas with a minimum breadth of 50 m. These gaps are irregular and often sinuous in form.

Ten replicate patches within each of the six patch stages (60 plots total) were chosen for the study, prioritizing patches that were not spatially auto-correlated with regard to patch stage and had an area of at least 90 m$^2$. Half were in the Typic Kandiudult soil subgroup (Faceville, Orangeburg, Fuquay series) and half in the Arenic Kandiudult soil subgroup (Lucy series) [44]. This stratification resulted in five replicate plots per patch stage and soil type combination (Figure 1A). Patches were further stratified to incorporate the different fire histories of the two burn units. The six patch stages were replicated twice in each soil type on the west burn unit and three times on the east burn unit. Patch areas ranged from 90 m$^2$ to 1045 m$^2$ (median 320 m$^2$). At the approximate center of each selected patch, a steel reinforcement bar was used to establish the center of a plot for sampling plants and physical conditions.

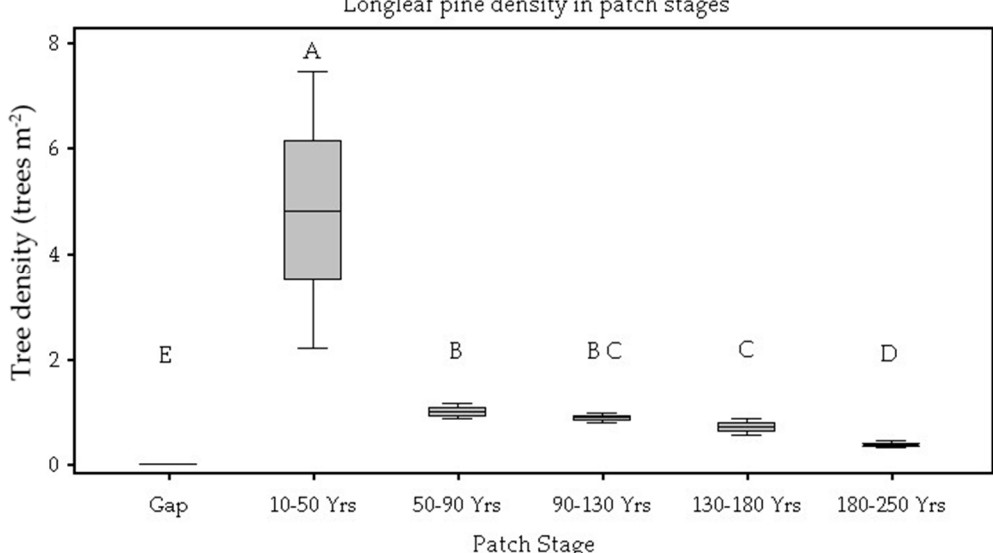

**Figure 1.** Mean longleaf pine tree density (trees m$^{-2}$) for plots in each patch stage (*n* = 10). Boxes represent plot distribution within one standard deviation of the mean (box centerline). Error bars indicate standard error and shared letters indicate non-significant differences between patch stages.

### 2.2. Ground-Layer Vegetation

We censused plant species in the ground-layer vegetation during the growing season from July through August in 2017. The census timing was staggered to be approximately four months following prescribed fires in each burn unit (east unit, March 23, 2017, west unit, April 12, 2017). Within a square 10 m$^2$ area around each plot center, we recorded a list of all vascular plant species <2 m tall and with stems ≤2 cm diameter at breast height (approximately 1.4 m). The nomenclature followed Weakley [45]. Using 10 cover classes, we also estimated relative aerial percent cover [46] of every species within this area. These estimates were performed by one observer to minimize bias among patch stages [47–49]. We also measured aboveground biomass (g) of broadleaf woody plants capable of forming a canopy over herbaceous plants to assess their potential interactions with trees and herbaceous plants. We measured the basal stem diameter (within 3 cm above the soil surface) of plants rooted within the 10 m$^2$ plot. We calculated their aboveground biomass (g) using locally determined allometric equations [30] to assess their potential competitive or facultative interactions with trees and herbaceous plants. Additionally, we approximated aboveground productivity of ground-layer plants since the previous prescribed fire by clipping live aboveground vegetation rooted within two 0.25 m$^2$ sampling frames placed within a 10 m radius of the plot center and sorted material according to its origin from graminoid, forb, or woody plants (<2 cm dbh and <2 m height). After drying and weighing the samples, we calculated the total biomass per unit area (kg m$^{-2}$) for each of the three categories.

### 2.3. Fire Behavior and Fuel Loads

We compared fuel loads, fuel consumption, and associated energy release during fire among the different patch stages by sampling biomass before and after the 2017 fires. Both units had been previously burned one year prior. Before each fire, we harvested all aboveground plant biomass and litter in two 0.25 m$^2$ subplots per plot and sorted it by herbaceous plants (live and dead graminoids and forbs), broadleaf woody material (live and dead broadleaf plant leaves and stems less than a 0.6 cm diameter), pine needle litter, woody pine litter (branches, bark, and pine cones sorted separately), and fine particulate matter (small pieces of litter including residual material from previous burns) before drying and weighing them.

Immediately after the fires, we re-sampled residual material within two more 0.25 m$^2$ subplots and sorted it into live biomass (<0.6 cm thickness), dead fine biomass (<0.6 cm thickness), and coarser

dead biomass (0.6–2.5 cm thickness). To estimate total fuel energy release per unit area, we used the sum of category-specific fuel energy contents [50] and applied it to the total fuel consumed.

### 2.4. Canopy Structure

We measured tree density and basal area ($m^2$ $ha^{-1}$) in plots by recording the diameter at breast height (DBH) of every tree within a 5-m radius of the plot center (78.5 $m^2$ area) in the 10–50, 50–90, and 90–130 years patches and, to accommodate for greater dispersion of trees, within a 10-m radius of the plot center (314 $m^2$ area) in the 130–180 and 180–250 years patches. We measured the percentage of canopy openness as an indicator of available light at each of the 60 plot centers using a digital camera with a hemispherical lens mounted on a tripod at 1 m height and oriented vertically to encompass only the canopy and not the ground-layer vegetation. Using the software Gap Light Analyzer, we calculated the percentage of open canopy as an index of available light.

### 2.5. Soil Characteristics

To characterize soil attributes that might influence plant growing conditions and species distributions, we measured soil bulk density, water content, total N and C, organic matter, pH, and concentrations of K, Ca, and Mg in each of the 60 plots. For soil chemical content, a 2-cm diameter soil probe was used to collect 10 soil cores to a 10-cm depth within a 5-m radius of the plot center, after which cores were combined for each plot. Soil samples were air-dried, pulverized, and filtered through a 2-mm sieve to remove fine roots and mineral concretions. These samples were processed at a soils laboratory at Auburn University, where soil mineral nutrients and acidity with an iCAP analyzer (Thermo Fisher Scientific Inc., Waltham, MA, USA) were measured. Organic matter, total N, and total C using a TruSpec CN analyzer (Leco Corps, St. Joseph, MI, USA) was also measured. Bulk density was measured using a slide hammer and soil collection barrel with a plastic insert to collect undisturbed cores to a 10-cm depth at three locations within a 3-m radius of each plot center. Samples were dried and weighed to calculate the mean soil bulk density (g $cm^{-3}$) per plot. Soil volumetric water content was sampled in both the summer rainy season (July) and the early dry season (October) by collecting five replicate soil samples to a 10-cm depth within 5 m of each plot center during a 12-h period. We calculated the per unit volume of soil volumetric water content (g $cm^{-3}$) at each plot location by subtracting the difference in the wet and dry masses of samples.

### 2.6. Data Analyses

We performed a two-factor univariate PerMANOVA (PC-Ord, MjM Software, Corvallis, OR) to test for differences among the patch stages for each univariate response variable measured, which includes the patch stand structure (tree density, basal area, canopy openness), soil attributes (N, C, K, Ca, Mg, pH, bulk density, water content, and organic matter), fire fuel loads (total and per each fuel category), total energy released during fires, and plant productivity (total and per each category). Soil subgroups were used as a blocking factor. PerMANOVA uses permutation tests to provide comparisons among patch stages [51] where differences were considered significant at $\alpha = 0.05$ [52].

To identify patterns in ground-layer plant species distribution among the patch stages, we conducted a non-metric multidimensional scaling (NMDS) analysis using plant species percent cover (midpoint of cover classes) within each plot as the response variable. We also conducted a series of indicator species analyses (ISA, PC-Ord, MjM Software, Corvallis, OR) to identify plant species, genera, and families that were significantly associated with particular patch stages or sets of combined patch stages. Given the possibility that species might be associated with multiple sequential stages, we performed an indicator species analysis using various combinations of the original six stages, where gaps were considered to precede the 10–50-year stage and follow the 180–250-year stage. Specifically, we created two new sets of stages by pairing sequential stages into three groups, and we created three additional sets by grouping three stages into two groups. Separate tests were run for each set for species, genera, and families to identify possible relationships between taxonomic groups and

the patch stage. Given the increased likelihood of finding significance for a taxonomic group using multiple analyses, "significant" results for individual species were considered descriptive for purposes of identifying general trends among genera and families. We related plant species composition to environmental characteristics of patches by performing a Canonical Correspondence Analysis (CCA) (PC-Ord, MjM Software, Corvallis, OR, USA) [51] using multiple measured environmental variables as explanatory variables and plant percent cover as the principal components' multivariate response variable. Prior to the CCA, we conducted a Principal Components Analysis (PCA, PC-Ord, MjM Software, Corvallis, OR, USA) and correlation analyses to identify environmental variables for use in the CCA to represent others that were highly correlated and logically related in order to not over-fit the model [51].

## 3. Results

### 3.1. Patch Environmental Characteristics

The structure and characteristics of longleaf pine patches generally followed patterns that were related to the regeneration cycle. Following the gap stage, the initially high density of trees in the 10–50 year old patches was followed by low densities that slowly decreased in the older patches (Figure 1). Average tree basal area increased as the ages of patches increased, with the highest average value in the 130–180 year category before declining as the clusters of trees forming patches dissipated (Figure 2). Canopy openness showed a roughly inverse pattern to tree density (Figure 1), with the highest values in gaps and the lowest values in the two youngest patch stages (Figure 3). Thus, the first century after pine colonization is when the lowest light conditions occur during the pine regeneration cycle.

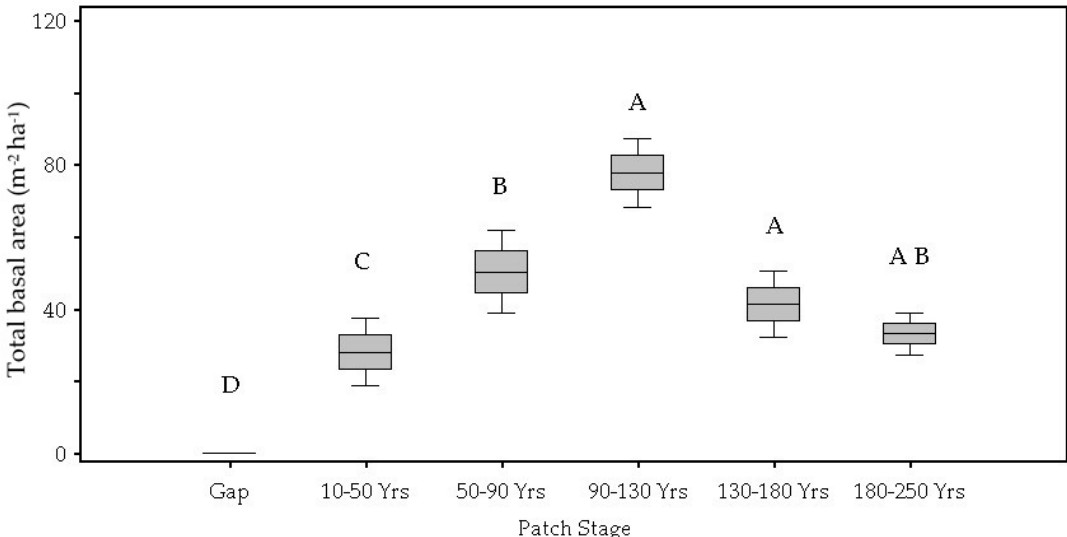

**Figure 2.** Total basal area in patch stages. Total basal area ($m^{-2}ha^{-1}$) of *Pinus palustris* for plots in each patch stage (*n* = 10). Boxes represent plot distribution within one standard deviation of the mean (box centerline). Error bars indicate standard error and shared letters indicate non-significant differences between patch stages.

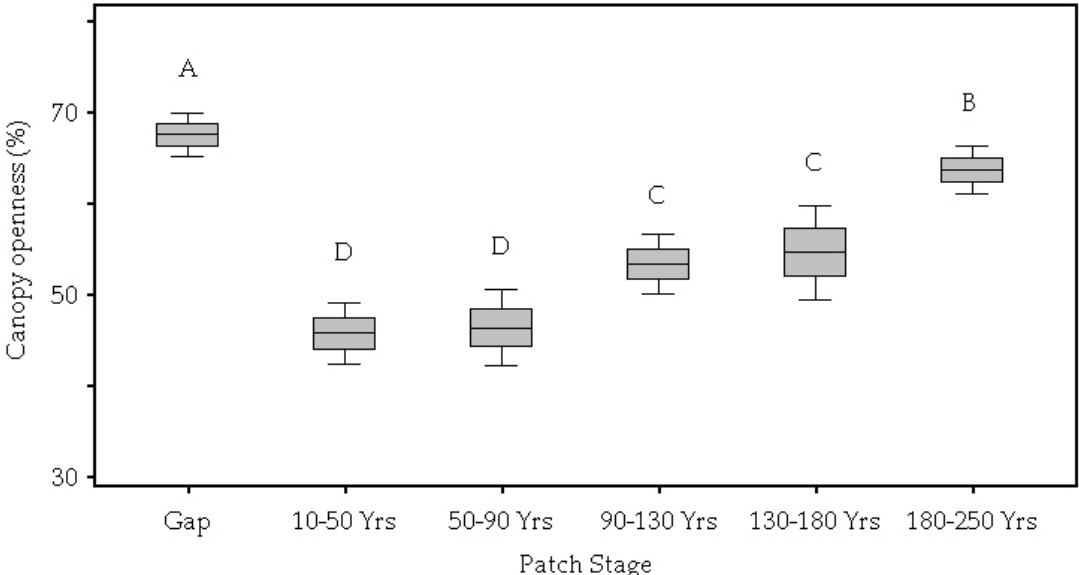

**Figure 3.** Total canopy openness in patch stages. Canopy openness (%) of plots in each patch stage (*n* = 10). Boxes represent plot distribution within one standard deviation of the mean (box centerline). Error bars indicate standard error and shared letters indicate non-significant differences among patch stages.

Other physical conditions also demonstrated trends related to the regeneration cycle. Total fuel loads, pine needle fuel loads, and total fuel consumed were lowest in gaps compared to occupied patches, peaked at 50–90 years, and then steadily decreased in progressively older patch stages (Figure 4, Appendix A). Total fuel energy released during fires also peaked at 50–90 years (Figure 5, Appendix A). There were no significant differences among patch stages for live herbaceous fuel, broadleaf woody plant stems, or broadleaf woody plant leaves (Appendix A). Levels of total soil carbon were at the lowest in gaps and young tree patches, peaked at 90–130 years, and then decreased at older patch stages (Figure 6). Soil mineral nutrients (Ca, K, Mg) generally increased over the life span of trees and were highest in 180–250 year patches, but lacked distinct associations across patch stages (Appendix A).

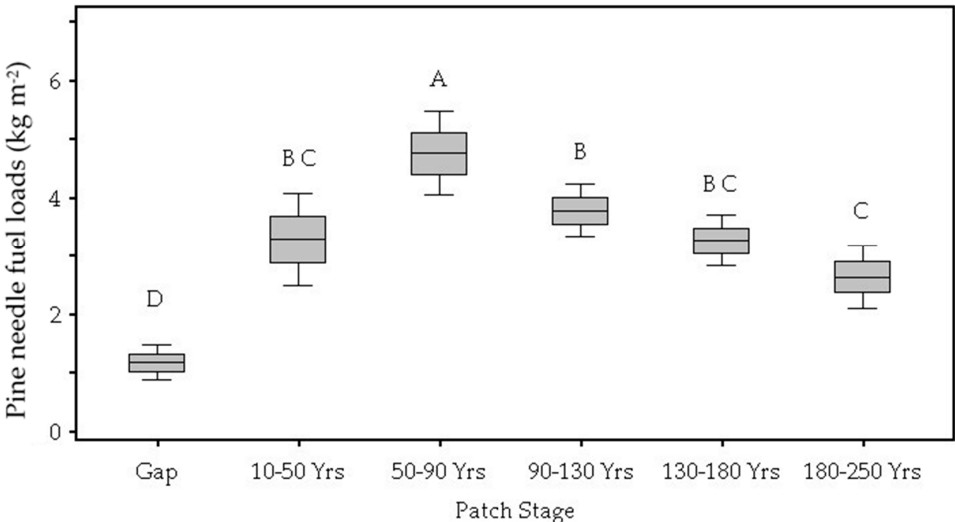

**Figure 4.** Pine needle fuel loads in patch stages. *Pinus palustris* needle litter accumulation (kg m$^{-2}$) in plots in each patch stage (*n* = 10). Boxes represent plot distribution within one standard deviation of the mean (box centerline). Error bars indicate standard error and shared letters indicate non-significant differences among patch stages.

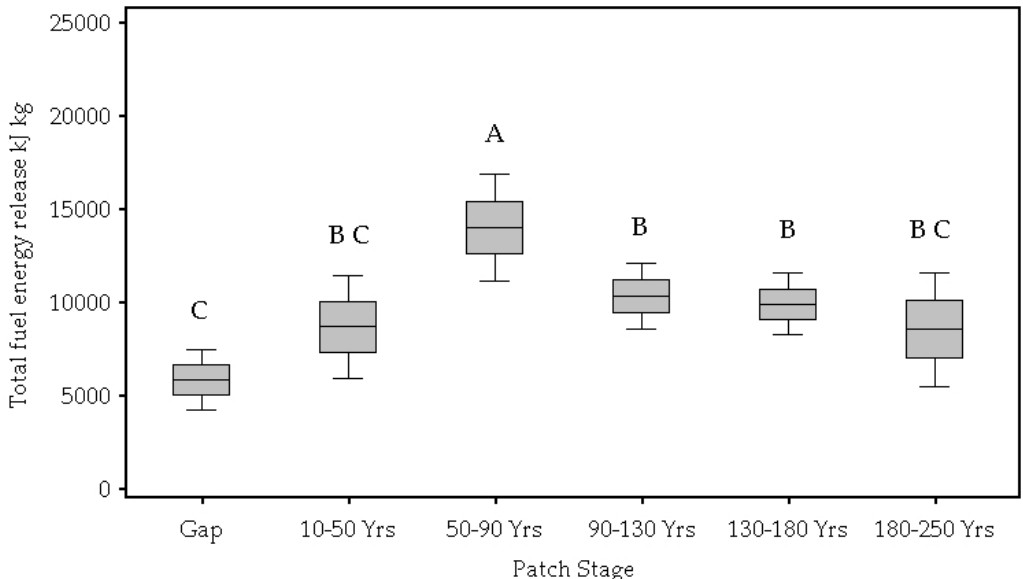

**Figure 5.** Total energy release in patch stages. Total energy release (kJ/kg) of fire fuel loads from the prescribed burn in March and April 2017 in plots in each patch stage (*n* = 10). Boxes represent plot distribution within one standard deviation of the mean (box centerline). Error bars indicate standard error and shared letters indicate non-significant differences among patch stages.

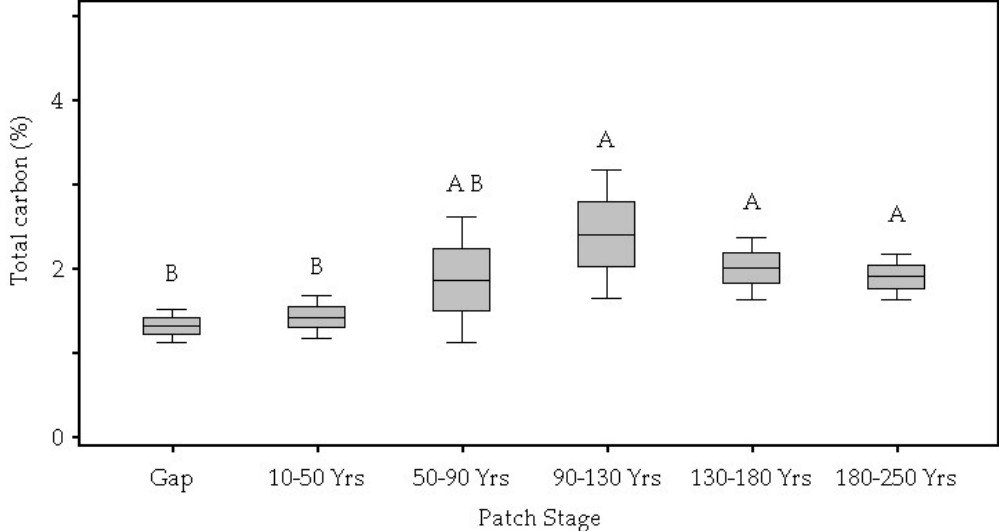

**Figure 6.** Total soil carbon in patch stages. Total percent of soil carbon in plots in each patch stage (*n* = 10). Boxes represent plot distribution within one standard deviation of the mean (box center line). Error bars indicate standard error and shared letters indicate non-significant differences among patch stages.

The PCA scatterplot reflected associations between patch stages and environmental variables identified in the univariate tests and showed how many variables were correlated with each other. Points representing plots are distributed in a roughly counter-clockwise cyclical arrangement in order of increases in patch age from gaps in the lower right through successive patch ages to the oldest patches in the lower center (Figure 7). Based on the PCA (Figure 7) and correlations among variables (Table 1), three variables were selected for the CCA. These were total basal area, pine needle litter, and canopy openness.

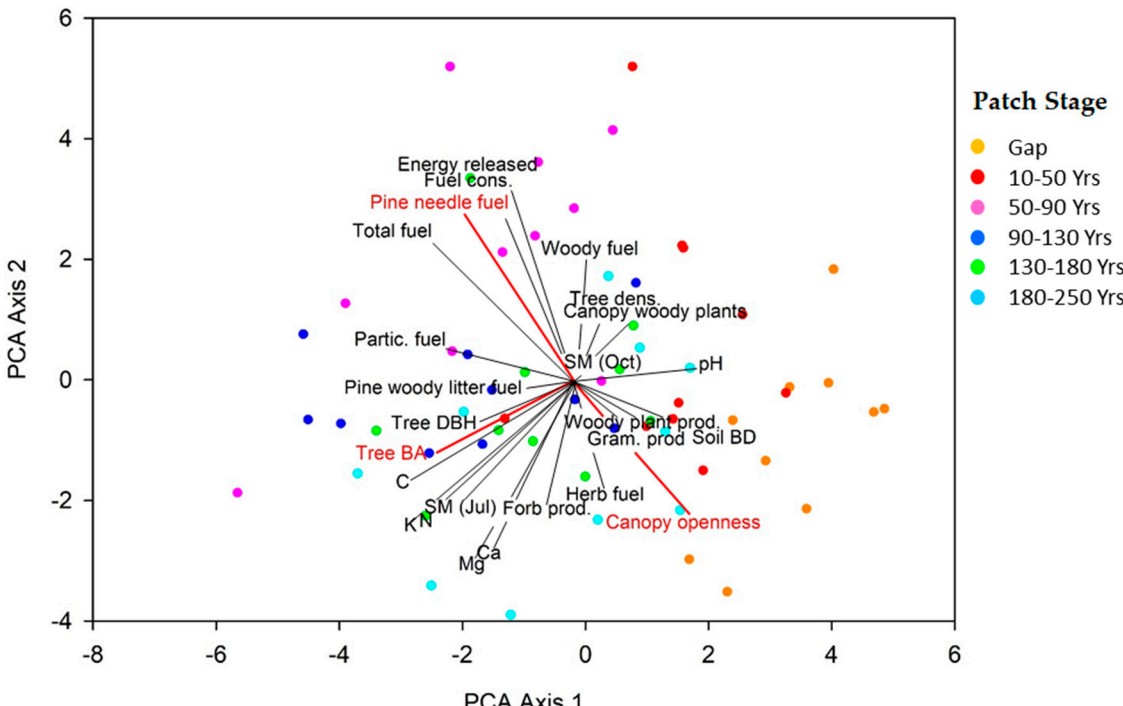

**Figure 7.** PCA ordination graph of the environmental characteristics in plots in each patch stage (*n* = 10). Each point represents a plot (*n* = 60). Soil characteristics include organic matter (OM), total nitrogen and carbon, pH, soil bulk density (SBD), and two soil moisture (SM) periods. Fire behavior attributes were of live fuel loads (herbs, woody stems) and deposited litter (woody leaves, pine needle litter, and non-needle pine litter). Measurements of plant productivity (graminoids, forbs, and woody biomass) and canopy structure (pine mean dbh and tree density) were also included. The names of the three environmental characteristics chosen for the CCA are highlighted in red: needle pine litter, canopy openness (light levels), and total basal area.

**Table 1.** Correlations (r = Pearson's correlation coefficient) of each patch environmental variable included in the PCA with the three representative variables used in the CCA.

| Canopy Openness | r | Pine Basal Area | r |
|---|---|---|---|
| Herbaceous fuel | 0.333 | Pine tree mean dbh | 0.819 |
| Graminoid productivity | 0.274 | Fine particulate fuel | 0.480 |
| Woody plant productivity | 0.268 | Pine woody litter fuel | 0.453 |
| Canopy woody plant biomass | 0.272 | % total C | 0.450 |
| Tree density | −0.482 | % total N | 0.370 |
| | | Ca ppm | 0.341 |
| Pine needle litter | r | Mg ppm | 0.364 |
| Total fuel | 0.706 | K ppm | 0.503 |
| Total fuel consumed | 0.657 | pH | −0.445 |
| Total energy released | 0.751 | Soil bulk density | −0.162 |
| Forb productivity | −0.225 | Woody plant fuel | −0.210 |
| Soil bulk density | −0.238 | Woody stems | −0.183 |

### 3.2. Plant Community Composition among Patch Stages

The NMDS analysis on plant species indicated considerable variability in plant species composition among the replicate plots within each patch stage (Figure 8). However, indicator species analyses identified certain plant species, genera, and families to be associated with particular patch stages or stage groupings. Plant species in the family Asteraceae were most strongly associated with gaps and 180 to 250-year-old patches (Table 2). Relatively high light and low nutrient environments characterized these patches (Figures 3 and 6). Five of the 24 species in Asteraceae had significant associations with

patch stage groupings that included gaps (*Chrysopsis mariana* Elliot, *Helenium* spp., *Eupatorium album* Michaux, *Helianthus radula* Torrey and Gray, *Verbesina aristata* Elliot) and an additional ten species in the family were the most abundant in patch stage groupings including gaps (Table 2). The family Asteraceae as a whole was significantly associated with the gap/10–50 year grouping, with 15 out of 24 species having the strongest associations with groupings that included gaps, even though a few species in the family had significant associations with non-gap stage groupings (*Ageratina aromatic* Spach, *Sericocarpus tortifolius* Michaux, *Symphyotrichum adnatum* Nesom, *Vernonia angustifolia* Michaux). Species in families Convolvulaceae (*Ipomoea pandurate* Meyer), Lamiaceae (*Scutellaria multiglandulosa* Small), and Rosaceae (*Rubus cuneifolius* Pursh) had significant associations with patch stage groupings including gaps (Table 2). The family Euphorbiaceae as a whole was similarly associated with the 180–250 year/gap/10–50 year stage grouping. While Poaceae as a whole did not have a significant association, the genus *Dichanthelium* was significantly associated with the gap/10–50 year/50–130 year grouping (Table 2).

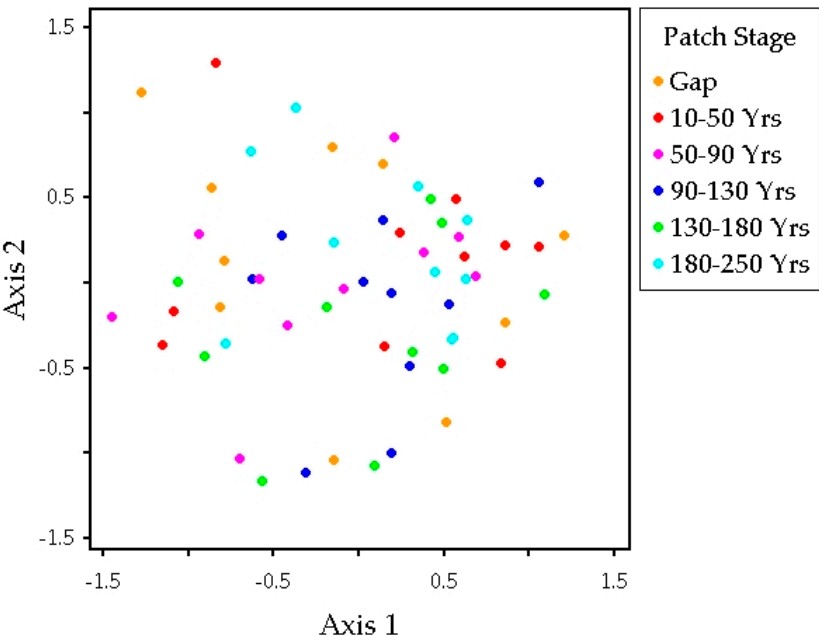

**Figure 8.** NMDS ordination graph of ground-layer plant species' relative cover in plots of each of the patch stages (*n* = 10). Each point is a plot of a patch, with color denoting its corresponding stage.

Some genera were related to patch stages representing intermediate age classes of trees. In the family Fabaceae, seven out of its 26 species (*Centrosema virginianum* Benth., *Desmodium ciliare* Muhl ex. Willd, *D. lineatum* Gray, *D. marilandicum* Kuntze, *Galactia volubilis* Britton, *Lespedeza angustifolia* Elliot, and *L. repens* Barton) were significantly associated with stage groupings incorporating the 90–130 year age class, while an additional 10 species had similar but non-significant associations (Table 2). The genera *Desmodium*, *Galactia*, *Lespedeza*, and *Tephrosia* as a whole also had significant associations with patch stage groupings that included the 130–180 year stage. None of the species in Fabaceae had significant associations with patch stage groupings that included gaps (Table 2).

Some plant taxonomic groups were associated with environmental variables characteristic of certain patch stages. Results from the CCA (*p* = 0.002) indicated that species in the family Fabaceae as a whole were associated with relatively lower light levels, higher levels of pine litter (and by correlation fuel load, consumption, and energy release), and high basal area (and, by correlation, total soil carbon, nitrogen, and mineral nutrients), (Figure 9). Species in Asteraceae and Rosaceae were related to relatively high light levels and low fuel loads, basal area, and nutrient levels (Figure 9). The genus *Dichanthelium* and family Euphorbiaceae were centrally distributed on the CCA graph with regard to fuel loading and canopy openness but were on the low side of the basal area (Figure 9).

**Table 2.** Indicator species analysis results including the patch stage or stage group with which species, genera, and families were most strongly associated. Taxonomic groups included are those with at least one significant result. Bold *p*-values indicate significant associations.

| Family | Genus | Species | Patch Stage Association (Years) | *p*-Value |
|---|---|---|---|---|
| Acanthaceae | *Dyschoriste* | *oblongifolia* | **10–130** | **0.009** |
| Asteraceae | | | **Gap–50** | **0.026** |
| | *Ageratina* | *aromatica* | **10–130** | **0.038** |
| | *Chrysopsis* | *mariana* | **Gap–50** | **0.015** |
| | *Conyza* | *canadensis* | 50–180 | 0.155 |
| | *Elephantopus* | | 130–Gap | 0.305 |
| | | *elatus* | Gap–50 | 0.274 |
| | | *tomentosus* | 90–180 | 0.104 |
| | *Eupatorium* | | **180–Gap–50** | **0.031** |
| | | *album* | **180–Gap–50** | **0.025** |
| | | *leucolepis* | Gap–130 | 1.000 |
| | *Helenium* | spp. | **Gap–50** | **0.031** |
| | *Helianthus* | *radula* | **Gap–50** | **0.010** |
| | *Hieracium* | *gronovii* | 130–Gap | 0.425 |
| | *Liatris* | *elegantula* | 180–250 | 1.000 |
| | *Pityopsis* | | 130–Gap | 0.206 |
| | | *aspera* | 130–Gap | 0.380 |
| | | *graminifolia* | 130–Gap | 0.275 |
| | *Rudbeckia* | *hirta* | 180–Gap–50 | 0.201 |
| | *Sericocarpus* | *tortifolius* | **130–180** | **0.043** |
| | *Solidago* | | 130–Gap | 0.197 |
| | | *altissima* | 130–180 | 0.170 |
| | | *odora* | 180–Gap–50 | 0.254 |
| | *Symphyotrichum* | | 90–180 | 0.124 |
| | | *adnatum* | **90–180** | **0.019** |
| | | *concolor* | Gap–50 | 0.539 |
| | | *dumosum* | 50–180 | 0.061 |
| | | *oolentangiense* | 130–Gap | 0.239 |
| | *Trilisa* | *odoratissima* | Gap–180 | 1.000 |
| | *Verbesina* | *aristata* | **180–Gap–50** | **0.018** |
| | *Vernonia* | *angustifolia* | **90–180** | **0.054** |
| Convovulaceae | | | **Gap–50** | **0.008** |
| | *Ipomoea* | *pandurata* | **Gap–50** | **0.004** |
| | *Stylisma* | *patens* | 130–Gap | 0.242 |
| Euphorbiaceae | | | **180–Gap-50** | **0.023** |
| | *Cnidoscolus* | *stimulosus* | 10–90 | 0.392 |
| | *Euphorbia* | | 10–90 | 0.466 |
| | | *discoidalis* | 10–90 | 0.226 |
| | | *heterophylla* | 180–250 | 1.000 |
| | *Stillingia* | *sylvatica* | 180–Gap–50 | 0.087 |
| | *Tragia* | | 180–Gap–50 | 0.340 |
| | | *urens* | 180–Gap | 0.190 |
| | | *urticifolia* | 10–130 | 0.512 |
| Lamiaceae | *Scutellaria* | *multiglandulosa* | **Gap–50** | **0.053** |
| Poaceae | *Dichanthelium* | | **Gap–90** | **0.007** |
| | | *aciculare* | 50–130 | 0.283 |
| | | *acuminatum* | Gap | 0.129 |
| | | *angustifolium* | Gap–90 | 0.241 |
| | | *dichotomum* | 130–250 | 0.613 |
| | | *ovale* | Gap–90 | 0.411 |
| | | *ravenelii* | 10–90 | 0.093 |
| | | *strigosum* | **Gap–90** | **0.017** |
| | | *villosissimum* | 10–90 | 0.778 |
| | *Gymnopogon* | *ambiguus* | **50–130** | **0.037** |
| Rosaceae | *Rubus* | | **Gap–50** | **0.010** |

**Table 2.** *Cont.*

| Family | Genus | Species | Patch Stage Association (Years) | *p*-Value |
|--------|-------|---------|--------------------------------|-----------|
| | | *cuneifolius* | **Gaps** | **0.045** |
| | | *flagellaris* | Gap–50 | 0.076 |
| Fabaceae | | | **50–130** | **<0.001** |
| | *Centrosema* | *virginianum* | **90–250** | **0.005** |
| | *Chamaecrista* | *fasciculata* | 130–Gap | 0.298 |
| | *Clitoria* | *mariana* | 10–130 | 0.218 |
| | *Crotalaria* | *rotundifolia* | 180–250 | 0.151 |
| | *Dalea* | *albida* | 10–50 | 1.000 |
| | *Desmodium* | | **90–130** | **0.002** |
| | | *ciliare* | **90–130** | **0.010** |
| | | *floridanum* | 130–250 | 0.283 |
| | | *glabellum* | 130–250 | 0.331 |
| | | *lineatum* | **90–130** | **0.002** |
| | | *marilandicum* | **130–180** | **0.020** |
| | | *obtusifolium* | 10–50 | 1.000 |
| | | *paniculatum* | 10–130 | 0.286 |
| | | *viridiflorum* | 50–130 | 0.092 |
| | *Galactia* | | **50–180** | **0.021** |
| | | *regularis* | 10–90 | 0.175 |
| | | *volubulis* | **50–130** | **0.013** |
| | *Lespedeza* | | **50–180** | **0.025** |
| | | *angustifolia* | **130–250** | **0.050** |
| | | *repens* | **10–130** | **0.030** |
| | | *virginica* | 130–Gap | 0.236 |
| | *Mimosa* | *quadrivalvis* | 50–180 | 0.430 |
| | *Strophostyles* | *umbellata* | 50–180 | 0.386 |
| | *Stylosanthes* | *biflora* | 50–180 | 0.271 |
| | *Rhynchosia* | *reniformis* | 50–180 | 0.412 |
| | | *tomentosa* | 50–180 | 0.286 |
| | *Tephrosia* | | **50–180** | **0.029** |
| | | *florida* | 50–180 | 0.348 |
| | | *spicata* | 50–180 | 0.492 |
| | | *virginiana* | **50–90** | **0.044** |

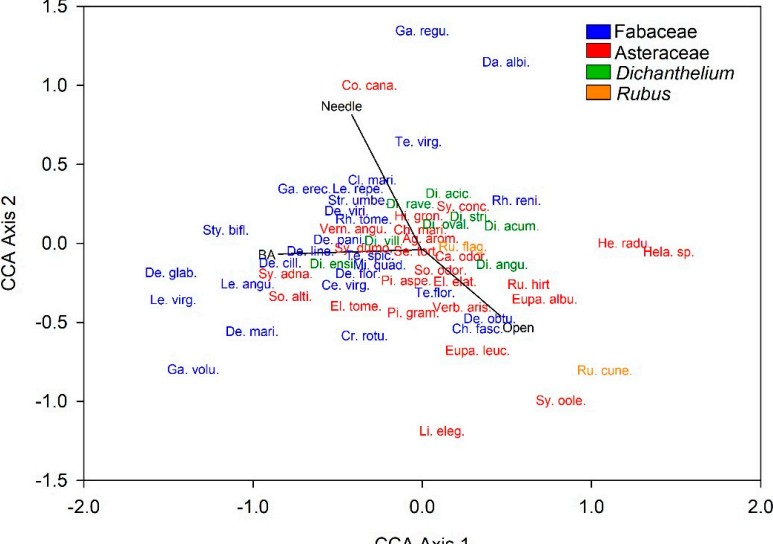

**Figure 9.** CCA ordination graph of the ground-layer vegetation genus or families' relative cover in relation to three environmental characteristics: pine litter accumulation, basal area, and canopy openness (light levels). Each point represents a species color-coded into a family or genus, as indicated by the legend.

## 4. Discussion

Although plant species were not clearly related to longleaf pine regeneration patch stages at the whole-community level, our study indicated that certain plant species, genera, and families were associated with certain stages of the regeneration cycle of longleaf pine. The strongest associations were for three families (Asteraceae, Fabaceae, and Euphorbiaceae) and the genera *Dichanthelium* and *Rubus.* These taxa collectively include about one-third of the known plant species on the Wade Tract. These associations suggest two broad patterns in trade-offs among life history characteristics that result in a tolerance of different stressors and use of different resources by suites of ground-layer plants.

Species in Asteraceae and genera *Dichanthelium* and *Rubus* were generally associated with patch stages characterized by relatively open over-story conditions represented by the oldest patches and gaps. Many species in the Asteraceae are considered relatively inflexible, light-demanding species, with physiological characteristics like thick, hairy leaves with waxy cuticles and extensive taproots that make them more resistant to water stress in high light environments [9], like those in which they most often occurred in this study. These species, as well as grasses in the genus *Dichanthelium*, also may be more sensitive to relatively high severity fire and, thus, tend to be most abundant in areas with minimal litter accumulation and heat release. *Dichanthelium* species have been found to be associated with younger longleaf pine age classes in other studies [53]. *Rubus cuneifolius*, which is a heliophilic species [54] strongly associated with gaps in our study, has perennial stems which, like broadleaf woody plants, gain a competitive advantage over herbs where fire severity is lower, such as in gaps. The species is relatively non-flammable and typically requires two years of aboveground growth before producing berries [54], such that sexual reproduction is most likely in gaps that are less frequented by fire.

Species in family Fabaceae were generally associated with intermediate patch ages. Many species in Fabaceae have been previously recognized as somewhat shade tolerant [55–57]. Most of these species were associated with patches that have high tree density, basal area, and canopy cover and have been observed occurring in similar conditions in other pine savanna systems [53]. Seeds of some species in Fabaceae have exhibited high heat tolerance during fires [58] and have fire-stimulated seed germination [59]. These conditions correspond to those in mid-age patches where species in Fabaceae were preferentially found. The ability of many legumes to maintain persistent seed banks [60], disperse seeds on the fur of mammals, and survive ingestion [61,62] may favor their occurrence in areas where increased fire severity causes death of genetic individuals, which requires a re-establishment through seed dispersal and germination instead of re-sprouting. The association of several species of Fabaceae with a high basal area and by association-high soil carbon corresponds to findings that many can increase soil carbon by stimulating belowground biomass and restoring soil N lost through volatilization in frequent fire [63] via nitrogen fixation [64], including the genera *Centrosema* and *Tephrosia* [65]. However, nitrogen was not found to vary significantly among patch stages.

Associations between certain taxa and patch stages notwithstanding, whole-community level associations were not evident. This lack of community-level variation apparently reflects considerable variation in species composition among plots within patch stages, such that species were not consistently represented in all plots within the stage. An under-sampling of the species composition using 10 m$^2$ plots may be one possible underlying reason, given that the median patch size was 320 m$^2$ and >500 species are present within the study area. It is particularly notable that species in the family Poaceae (except *Dichanthelium*) were not recognizably associated with stages of the tree regeneration cycle, given their high richness (ca. 20% of total species) and physical dominance in the herbaceous layer. Their robust presence, despite large variation in tree cover and pine fuels among patches of pines on the Wade Tract, presumably enhances fire spread and contributes to the overall high fire frequency and low broadleaf woody plant dominance that characterizes pine savannas [15,20,66,67].

Our study was the first to quantitatively characterize stages of the longleaf pine regeneration cycle in terms of the tree population structure, fuel load and consumption, and soil characteristics. Early stages were characterized by higher rates of mortality [11,24], presumably due to density-dependent

competition and fire damage to younger, relatively more susceptible trees [11,25,26]. The peak of needle litter accumulation, fuel consumption, and energy release during fires during middle stages indicated that increasing needle productivity per tree overrides effects of decreasing tree density on needle fall as the patches age [25]. Needle fall diminishes in older patches and gaps, where there are fewer to no trees to deposit litter and, therefore, fire severity is lower [24–26]. As a consequence, fuel loads, energy release, and associated fire severity are inversely related to light availability over the course of the regeneration cycle. Certain soil characteristics also appear to be mediated by longleaf pine patch dynamics. Soil mineral nutrients (Ca, K, Mg), which were highest in the oldest tree size classes, increase after fire [63]. This pattern suggests that, in this study, these nutrients accumulated because of increasing fuel consumption and deposition of residual ash. The relatively low soil bulk density near the middle patch stage of trees was possibly because the fine roots of competing trees increased soil porosity [68]. The peak in organic matter and total soil carbon in the 50–180 year patch stage corresponds to the highest rates of potential biomass addition in the form of pine needles. Previous studies have shown turnover rates of organic matter originating from pine needles to be slower than that from other types of litter, which results in organic matter accumulation [69]. Higher C:N ratios in soil of the intermediate aged patches might slow decomposition rates [69]. Pine needles typically have much lower N concentrations than other litter sources even though the effect on decomposition rates may vary [70–72]. Pine needles also tend to char more than litter from broadleaf woody plants and herbaceous plants (authors' observations), which could result in the introduction of more recalcitrant carbon compounds to the soil [73]. Such changes in soil nutrients might potentially mediate some of our observed changes in the ground layer vegetation during the pine regeneration cycle.

## 5. Conclusions

Our results expand the conceptual model of a longleaf pine regeneration cycle in old-growth longleaf pine savanna by demonstrating that different stages have different environments that limit species distribution and promote spatial patterns in some plant taxonomic groups. Longleaf pine population dynamics, thus, contribute both to predictable spatial heterogeneity in local ground-level and soil environments, as well as in the ground-layer plant community. Although the pine regeneration cycle does not produce distinct species associations of ground-layer vegetation, its direct and indirect effects contribute to overall high plant species richness of the old-growth pine savanna ecosystem. It is possible that further sampling of the ground-layer vegetation in larger plots may further clarify these patterns with patch structure and age. This study was an important contribution toward understanding the environmental conditions that most influence the longleaf pine savanna structure and their interactions with ground-layer vegetation. Identifying these associations and the cyclic nature of the longleaf pine regeneration is key for informing restoration efforts that emulate the spatial and temporal patterns that characterize these communities. Future work in such rare, old-growth longleaf pine savannas will contribute to long-term data sets and use as a reference model for land management.

**Author Contributions:** Conceptualization, M.P.M., K.M.R., and D.L.M. Methodology, M.P.M., K.M.R., and D.L.M. Census dataset and tree age data, W.J.P. Field data collection, M.P.M. and K.M.R. Writing—review and editing, M.P.M., K.M.R., D.L.M., and W.J.P. Funding acquisition, K.M.R., McStennis, and D.L.M.

**Funding:** The Department of Wildlife Ecology and Conservation at the University of Florida and Tall Timbers Research Station funded this research. The long-term study that formed the basis for this current study has been funded by Tall Timbers Research, Inc., with different phases funded through the National Science Foundation by individual and collaborative research awards (BSR 8012090 & 8605318, W.P., PI; BSR-8718803 & 8718993, W.P. & J.H. PIs; DEB-8907138, W.P., PI; DEB 0950347 & 0950302, W.P. & B.B., PIs; NSF DEB 1557000 & 1556837, W.P. & B.S., PIs).

**Acknowledgments:** This work would not have been possible without the financial and academic support of the Department of Wildlife, Ecology and Conservation at the University of Florida, (McStennis, FLA-WFC-005853) and Tall Timbers Research Station. Thank you to Ashlynn Smith for assisting with field sampling and the Tall Timbers Research Station for lending us their field sampling equipment.

**Conflicts of Interest:** The authors declare no conflict of interest. The funders had no role in the design of the study, in the collection, analyses, or interpretation of data, in the writing of the manuscript, or in the decision to publish the results.

# Appendix A

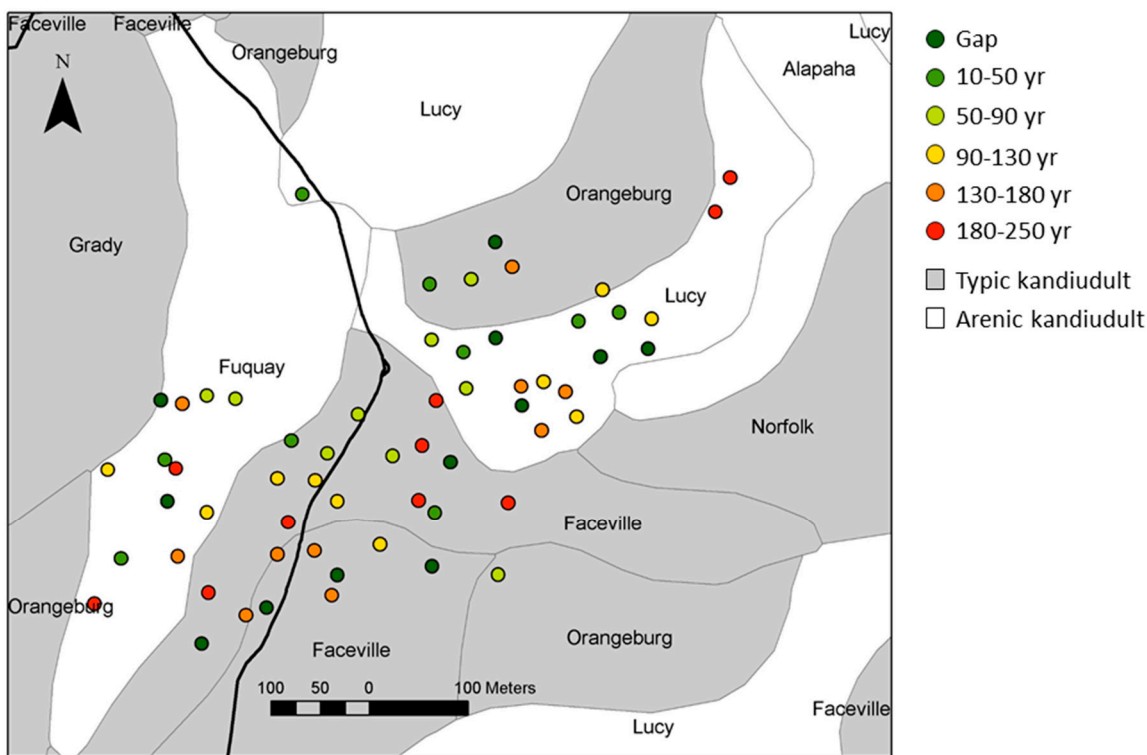

**Figure A1.** Map of the study site at the Wade Tract in Thomasville, GA. The points represent the locations of the selected study plots for each of the patch stages color-coded on the legend and their distribution on the east and west burn units and Arenic and Typica Kanduidult soil types.

**Table A1.** Mean values of different longleaf pine population structure attributes in each of the six patch stages and overall stage effect *p*-value from PerMANOVA analyses. Bold *p*-values indicate significant results ($p < 0.05$) and shared letter subscripts denote non-significant differences indicated by PerMANOVA pairwise comparisons.

| | | Patch Stage Mean | | | | | |
|---|---|---|---|---|---|---|---|
| Stage | | 0 | 1 | 2 | 3 | 4 | 5 |
| Analysis Factor | Overall *p*-Value | Gaps | 10–50 Years | 50–90 Years | 90–130 Years | 130–180 Years | 180–250 Years |
| Tree DBH (cm) | **<0.01** | 0.00 [E] | 11.88 [D] | 30.45 [C] | 41.15 [B] | 47.77 [B] | 59.95 [A] |
| Total Basal Area ($m^2$ ha) | **<0.01** | 0.00 [D] | 0.04 [C] | 0.08 [B] | 0.12 [A] | 0.13 [A] | 0.10 [A,B] |
| Tree Density (trees/10 $m^2$) | **<0.01** | 0.00 [E] | 4.82 [A] | 1.00 [B] | 0.88 [B,C] | 0.70 [C] | 0.37 [D] |
| Canopy Openness (%) | **<0.01** | 67.49 [A] | 45.69 [D] | 46.30 [D] | 53.27 [C] | 54.55 [C] | 63.62 [B] |

**Table A2.** Mean values of different soil attributes in each of the six patch stages and overall stage effect *p*-value from PerMANOVA analyses. Bold *p*-values indicate significant results ($p < 0.05$) and shared letter subscripts denote non-significant differences indicated by PerMANOVA pairwise comparisons.

| | | Patch Stage Mean | | | | | |
|---|---|---|---|---|---|---|---|
| Stage | | 0 | 1 | 2 | 3 | 4 | 5 |
| Analysis Factor | Overall *p*-Value | Gaps | 10–50 Years | 50–90 Years | 90–130 Years | 130–180 Years | 180–250 Years |
| Total Carbon (%) | **0.05** | 1.32 [B] | 1.42 [B] | 1.86 [A,B] | 2.40 [A] | 2.00 [A] | 1.90 [A] |
| Total Nitrogen (%) | 0.16 | 0.06 | 0.07 | 0.07 | 0.09 | 0.08 | 0.08 |
| Calcium (ppm) | **0.05** | 601.20 [B] | 613.20 [A,B] | 521.80 [B] | 776.40 [A] | 708.30 [A,B] | 899.60 [A] |
| Potassium (ppm) | **0.04** | 14.00 [B] | 16.00 [B] | 18.50 [B] | 19.70 [A] | 20.30 [A,B] | 22.90 [B] |
| Magnesium (ppm) | 0.10 | 104.70 | 121.10 | 105.40 | 141.60 | 131.10 | 153.90 |
| Soil bulk density (g cm$^{-3}$) | **0.02** | 0.5 [A,C] | 0.54 [A] | 0.44 [B] | 0.46 [B,C] | 0.48 [A] | 0.49 [A] |
| July soil water content (g cm$^{-3}$) | 0.42 | 0.22 | 0.23 | 0.24 | 0.26 | 0.26 | 0.24 |
| October soil water content (g cm$^{-3}$) | 0.45 | 0.28 | 0.32 | 0.24 | 0.24 | 0.24 | 0.25 |
| Soil pH | **0.02** | 5.79 [A] | 5.84 [A] | 5.60 [A] | 5.64 [B] | 5.62 [B] | 5.64 [A] |

**Table A3.** Mean values of different fire fuel load attributes (BL = broadleaf) in each of the six patch stages and overall stage effect *p*-value from PerMANOVA analyses. Bold *p*-values indicate significant results ($p < 0.05$) and shared letter subscripts denote insignificant differences indicated by PerMANOVA pairwise comparisons.

| | | Patch Stage Mean | | | | | |
|---|---|---|---|---|---|---|---|
| Stage | | 0 | 1 | 2 | 3 | 4 | 5 |
| Analysis Factor | Overall *p*-Value | Gaps | 10–50 Years | 50–90 Years | 90–130 Years | 130–180 Years | 180–250 Years |
| Total fuel loads (kg m$^{-2}$) | **0.002** | 0.52 [D] | 0.69 [C,D] | 1.06 [A] | 0.90 [B] | 0.82 [B,C] | 0.76 [C] |
| Total fuel consumed (kg m$^{-2}$) | **0.002** | 0.32 [C] | 0.46 [B,C] | 0.74 [A] | 0.54 [B] | 0.52 [B] | 0.45 [B,C] |
| Total energy released (kJ kg) | **0.00** | 5823.53 [C] | 8647.13 [C,B] | $0.13 \times 10^5$ [A] | $0.10 \times 10^5$ [B] | 9861.17 [B] | 8520.61 [B,C] |
| Total pine litter (kg m$^{-2}$) | **0.002** | 0.13 [D] | 0.35 [B] | 0.54 [A] | 0.39 [B,C] | 0.38 [B] | 0.27 [C] |
| Pine needle litter (kg m$^{-2}$) | **0.002** | 0.12 [D] | 0.33 [B,C] | 0.47 [A] | 0.38 [B] | 0.33 [B,C] | 0.26 [C] |
| Woody pine litter (kg m$^{-2}$) | **0.041** | 0.01 [B] | 0.03 [A,B] | 0.07 [A] | 0.05 [A] | 0.04 [A] | 0.03 [A] |
| Total BL woody (kg m$^{-2}$) | 0.19 | 0.15 | 0.11 | 0.18 | 0.06 | 0.09 | 0.11 |
| BL woody stems (kg m$^{-2}$) | 0.13 | 0.08 | 0.10 | 0.14 | 0.05 | 0.05 | 0.06 |
| BL woody leaves (kg m$^{-2}$) | **0.041** | 0.07 [A] | 0.03 [A,B] | 0.06 [A] | 0.02 [B] | 0.03 [A,B] | 0.03 [A,B] |
| Live herbs (kg m$^{-2}$) | 0.76 | 0.11 | 0.09 | 0.09 | 0.10 | 0.09 | 0.12 |
| Fine particulate litter (kg m$^{-2}$) | **0.002** | 0.13 [A] | 0.16 [A] | 0.25 [B] | 0.30 [B] | 0.27 [B] | 0.23 [B] |

**Table A4.** Mean values of estimated plant productivity, broadleaf (BL) woody plant biomass, and ground-layer plant community relative cover attributes in each of the six patch stages and overall stage effect *p*-value from PerMANOVA analyses ($p < 0.05$).

| | | Patch Stage Mean | | | | | |
|---|---|---|---|---|---|---|---|
| **Stage** | | **0** | **1** | **2** | **3** | **4** | **5** |
| **Analysis Factor** | **Overall *p*-Value** | **Gaps** | **10–50 Years** | **50–90 Years** | **90–130 Years** | **130–180 Years** | **180–250 Years** |
| Graminoid plant productivity ($\text{kg m}^{-2}$) | 0.42 | 0.03 | 0.02 | 0.02 | 0.03 | 0.03 | 0.03 |
| Forb plant productivity ($\text{kg m}^{-2}$) | 0.83 | 0.03 | 0.02 | 0.03 | 0.02 | 0.03 | 0.03 |
| BL woody plant productivity ($\text{kg m}^{-2}$) | 0.26 | 0.03 | 0.02 | 0.01 | 0.01 | 0.02 | 0.02 |
| BL woody plant biomass (g) | 0.26 | 584.67 | 299.13 | 404.58 | 181.72 | 220.68 | 672.51 |
| Canopy-closing woody Species' relative cover | 0.46 | 1.64 | 1.28 | 1.22 | 0.85 | 0.53 | 1.95 |
| BL woody species' relative cover | 0.47 | 1.59 | 1.49 | 1.34 | 1.58 | 1.52 | 2.08 |
| Total percent of plant coverage in plots | 0.17 | 97.70 | 104.21 | 90.90 | 116.70 | 98.43 | 98.35 |
| Ground-layer plant species richness | 0.32 | 36.70 | 41.60 | 36.90 | 39.20 | 37.00 | 34.20 |

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
