# Peer review of "Longleaf Pine Patch Dynamics Influence Ground-Layer Vegetation in Old-Growth Pine Savanna"

_forests, doi:10.3390/f10050389_

Round 1

Reviewer 1 Report

This paper explores how the physical processes related to the cycle of longleaf pine regeneration affect species composition in a relic old-growth forest.  The authors established 10 replicate plots for each of 6 patch stages: half in each of two soil subgroups.  Patch stages were identified by age categories: 10-50, 50-90, 130-180, 180-250, and treeless gaps.  Two sets were on one burn unit; the other three were on another burn unit.  The sample plots were 10m2.  The authors measured species richness and cover by species, as well as soils attributes and forest structural attributes.

I have three major concerns about the study, plus a number of other comments:

1)      Using tree age patches, soil types and burn unit to stratify plot locations sounds like a good approach to distribute the plots across the gradients in the study area.  However, more detail is needed here.  Did you map all the tree age patches, and then randomly select plots within each age group?  A map showing the age patches and plot locations would be useful.

2)      The shape of the sample plot was not specified.  It was also unclear how “species composition” was estimated, and how cover of individual species were estimated (see detailed comments below).  Were estimates of tree densities and basal areas all made in the 10m2? Further, as you correctly point out in the Discussion, 10m2 plots may be too small to adequately sample species composition.  Perhaps you can provide suggestions in the conclusion on what you feel would be a more appropriate plot size to better capture understory species composition and tree structural attributes? 

3)      It appears that a combination of small plot sizes and heterogeneous conditions within the age classes confounds the interpretation of the results.  This became very clear in Table 1, where, for example, the interpretation is that several species of Asteraceae are associated with gaps and therefore are light-loving.  However, all 5 of these species are also associated with the 10 to 50-year-old class, which has the highest tree density and the lowest openness. I am not seeing a clear preference for one class or another, but rather plants apparently occupying microsites within more than one age class.  A better approach would be to directly compare the cover of individual species to the attributes you measured instead of inferring relationships based on differences among tree age patches.

Other comments:

Lines 31-33:  I don’t think the statement that “changes in composition and structure of pine patches contributes to…predictable species composition.”  See 3) above.

Line 133-35:  The use of ring counts to estimate tree age has been shown to be inaccurate in some forest types, especially where missing and false rings are common.  I was just curious if you cross-dated even a subset of tree cores to provide some validation of the relative accuracy of using ring counts?

Lines 162-163 states that the sampling area was 10m2.  It is unclear if the 10m2 plot was circular, and if it was surrounded by a buffer of similar habitat.  It also states “we recorded species composition;” do you mean you estimated species richness by recording each species within the 10m2 plot?  If so, how did you do this (e.g., by subdividing the plot into sections and carefully searching each one? And, did you have more than one observer do this to estimate sampling error?)   On line 164, it states that you “estimated relative cover of all vascular plants…” I’m guessing that you estimated actual (not relative) aerial cover by species, since you referred to Daubenmire (1966)?  But, Daubenmire (1966) doesn’t specify how plant cover was estimated.  Did you make one estimate of the overall cover of each species in the 10m2 area?  Or, perhaps you instead meant to refer to Daubenmire (1959, NW Science 33:43-46), which specifies the use of 20 X 1250px quadrats and uses cover classes to estimate plant canopy cover?  Is that the technique you used in the field?  And, if so, how many quadrats did you use per plot and how were they distributed (e.g., every 1m along transects? randomly?)?

Lines 166-171: So, the basal areas and tree densities reported in the paper are based on estimates from the 10m2 plots? 

Line 217-218:  Sampling soil moisture is problematic, because it is difficult to get samples from all plots within a relatively short time period when no precipitation occurs.  Were you able to sample all your plots within a 12-hour period during which no precipitation occurred?

Fig. 2: basal area should be m2ha-1

Line 333, 368:  you refer to fig. 6 when discussing “nutrients,” but fig. 6 is actually just carbon

Line 363: Instead of “α”, I think this should be “P”

Table 1 and Fig. 9:  the line numbers switch to the right side for some reason and obscure the text

You could make a better case for why this study is important (e.g., old growth longleaf pine forests are pretty uncommon and we don't know a lot about how spatial patterns influence understory plant communities).  There also might be other relevant papers; here’s two that might be of interest, and there might be others:

Hammond et al. 2016. Forest Ecology and Management 364:154-164

Harrington and Edwards. 1999. Canadian J. Forest Research 29:1055-1064

Author Response

Attached as a Word document.

Reviewer 2 Report

This study tests whether the structural stage of longleaf pine stands influences ground layer dynamics and composition at species, genera, family and community levels. It leverages data from a vast chronosequence across an old-growth longleaf pine savanna in northern Florida/southern Georgia. The authors found that certain plant species, genera and families were associated with stages of longleaf pine stand development, but no evidence of community-level associations related to stage was found.

Overall, I found the manuscript to be generally well-written and the execution of the analysis to be sound. It is my opinion that this manuscript could merit publication in Forests if a few major revisions were made. First, there is a need for a sharpened hypothesis/objective. The specifics of the research question were not clear to me until after reading through the discussion section. Second, and related to the first point, the paper has organizational flaws that hinder the clarity of the message to the reader. Finally, further effort should be made to make this study more clearly relevant/impactful to the broad readership of Forests. Specific suggestions to improve the manuscript in light of these issues, as well as some minor issues, are detailed as follows.

L15-18 more succinct and clear to replace “We used an old growth longleaf pine savanna to test the conceptual model that physical processes…” with: “We asked, ‘do physical processes…in an old-growth pine savanna?’ Also I would find it helpful if ‘physical processes’ were defined more specifically here in the abstract.

L26-31 I recommend starting with what you did find at family level, then stating “however, community-level analyses did not…”

L48 change “could” to “can”

L50 change “should be especially likely” to “are likely”

L53 this hypothesis feels out of place (move to last paragraph of introduction section), and also needs to be more specific. It would be strengthened if you add “by…” to the end of the sentence. You can also be more specific than “patch dynamics”. I think you can say “developmental stage” here, no? The presentation of the hypothesis would also be improved with an alternate hypothesis. What would you expect to find if your hypothesis were proven wrong? Perhaps that ground layer dynamics are more associated with broader biophysical/environmental variables than they are with developmental stage.

L60 wording: replace “blocks” with “prevents”

L64 wording: replace “when applied” with “those that occur”

L64-65 very confusing phrase: “fires…have been designated an old-growth attribute of this community type” ??? please rephrase and clarify

L66-67 I do not think “patch dynamics” means what you think it means here. Please be more specific. As it is, this sentence sounds like circuitous reasoning.

L80 word choice: change “dissipate” (here and throughout text). Do you mean “become less defined” or “become smaller” perhaps?

L88 …is through fire severity…

L89 is “fire fuel” supposed to be “fine fuel”?

L90 change “resulting in” to “contributing to”

L97-102 Now is the place to state your hypothesis, and be specific. By “physical characteristics” do you mean light levels, soil moisture, soil N and C? What about fuels? Structural attributes? Throughout the paper, it appears you are using “environmental characteristics” and “physical characteristics” interchangeably, but there is never an entire list given of what you are referring to. If there is a distinction, make it explicit here. If there is no distinction, choose either “physical” or “environmental” and use it consistently.

Again, I think an alternate hypothesis would be valuable to set up how you will know whether you’ve answered your research question.

L102-104 remove this from this section; it belongs in the discussion.

L122 remove “during the 3.5 decades”

L123 does “timing” refer to interval between fires, or season of burning?

L125-126 replace “studies of fire scarred tree rings in the region going back for” with “the last”

L141-143 I still find the term “dissipate” confusing. Also, I don’t think you mean “without trees”, because you’re talking about a savanna-like matrix, correct? Some trees, but mostly open? Perhaps “with sparse tree cover” (or just leave simply as “open areas”)?

L145 why circles? Were the circles actually used as the sixth stage, or were irregular patches used as the sixth stage when they could contain a circle of 25-m radius?

L149 how were patches selected? Randomly stratified?

L223 I think this would be more clear to replace the period after “measured” with a colon and remove “we analyzed”. Is that what you mean to say here?

L251 the procedure is entirely unclear here. Did you use PCA only to determine which variables were collinear- if so, why not test for collinearity with pearson’s? Or did you actually run PCA and then use the principle components in further analysis? If this is the case, further explanation is needed. PCA doesn’t just reduce the number of variables by eliminating redundancies- it reduces dimensionality by combining the explanatory power of multiple variables along a single (novel) axis. Please be more clear.

L325-326 move this to discussion section

L329-330 Please be careful to be very clear when you are stating that certain species and families were associated with developmental stages, but that communities were not. This is not well-explained until L533 and it makes it sound as though you are contradicting yourself.

L579 I am left thinking “So what?”. Further effort should be made to relate this study to the broader discipline of forest ecology and management.

Fig 5 Is something wrong here? The 10-50 and 50-90 stage classes look to be indistinguishable, yet are demarcated as “BC” and “A”, respectively.

Fig 6 change title to Total soil carbon in patch stages. Also, for all figs, titles should be removed from top and placed at the start of the figure caption (like you’ve done with Fig 7).

Table A1 I would find it easier to read this (and all tables and figures throughout) if the letter subscriptions were in order from left to right. The first grouping from the left should be A, then the next new grouping would be B, etc.

Author Response

Word

Round 2

Reviewer 1 Report

This version of the manuscript addresses many of my previous comments and suggestions.  The interpretation of the results is still constrained by the combination of small plot sizes and heterogeneous conditions within the age classes.  However, these results provide hypotheses to be tested in future studies.  I noted a few edits:

line 51-53: this sentence doesn't make sense

line 173: delete "were" after "times"

Table 2: the line numbers obscure the text for some reason

line 529: delete "was generally"

Reviewer 2 Report

Thank you, the revisions that have been incorporated make the manuscript worthy of publication in Forests, in my opinion.

Author Response

Dear Reviewer,

Thank you for your feedback and your endorsement of our manuscript for publication in Forests. We are pleased that you found this version improved and appreciate your efforts and insight during the peer review process.

Thank you!

Sincerely,

Maria Paula Mugnani

Kevin Robertson

Deborah Miller

William Platt